# Exploring the Relationship between Antioxidant Enzymes, Oxidative Stress Markers, and Clinical Profile in Relapsing–Remitting Multiple Sclerosis

**DOI:** 10.3390/antiox12081638

**Published:** 2023-08-19

**Authors:** Anna Bizoń, Justyna Chojdak-Łukasiewicz, Sławomir Budrewicz, Anna Pokryszko-Dragan, Agnieszka Piwowar

**Affiliations:** 1Department of Toxicology, Faculty of Pharmacy, Wroclaw Medical University, Borowska 211, 50-556 Wrocław, Poland; agnieszka.piwowar@umw.edu.pl; 2Department of Neurology, Faculty of Medicine, Wroclaw Medical University, Borowska 213, 50-556 Wrocław, Poland; justyna.chojdak-lukasiewicz@umw.edu.pl (J.C.-Ł.); slawomir.budrewicz@umw.edu.pl (S.B.); anna.pokryszko-dragan@umw.edu.pl (A.P.-D.)

**Keywords:** relapsing–remitting multiple sclerosis, oxidative stress, antioxidants

## Abstract

We aimed to investigate the extent of alterations in the pro/antioxidant balance in the blood of patients with relapsing–remitting multiple sclerosis (RRMS) in relation to drug-modified therapy, gender, disability score, and disease duration. 161 patients (67 men and 94 women, aged 24–69 years, median 43.0) and 29 healthy individuals (9 men and 20 women, aged 25–68 years, median 41.0) were included in the study. We measured the activity of superoxide dismutase (SOD), glutathione peroxidase (GPx), and catalase (CAT) as well as the concentration of interleukin-6 (IL-6), lipid peroxidation parameters (LPO), total oxidant status (TOS), and total antioxidant capacity (TAS). The activity of SOD did not show any significant differences between patients with RRMS and the control group in our study. In contrast, significant decreased GPx activity and increased CAT activity was observed in the blood of patients with RRMS compared to the control group. Additionally, the activity of CAT was influenced by gender and the use of disease-modifying therapies. Disease-modifying therapies also affected the concentration of TOS, TAS, and LPO. Our studies indicated that enhancing GPx activity may be more beneficial to providing potential therapeutic strategies aimed at modulating antioxidant defenses to mitigate oxidative stress in this disease.

## 1. Introduction

Multiple sclerosis (MS) is a chronic inflammatory disorder of the central nervous system (CNS) characterized by demyelination, neurodegeneration, and variable clinical manifestations [1]. Although the exact etiology of MS remains elusive, much evidence suggests that oxidative stress and the disorders in prooxidant–antioxidant balance play a relevant role in the patomechanisms of this disease, especially with regard to the neurodegenerative component of its background. The CNS is highly susceptible to oxidative stress because of its high oxygen utilization, abundant lipid content, and limited ability to regenerate [2]. Additionally, oxidative stress affects neurons and all types of glial cells, but neurons and oligodendrocytes are the most sensitive to its impact [3]. Oxidative stress occurs when there is an imbalance between the production of free radicals and the body’s antioxidant defense system. This imbalance results from the excessive generation of reactive oxygen species (ROS) like the superoxide anion (O_2_^●−^), hydroxyl radicals (OH^●^), and hydrogen peroxide (H_2_O_2_). O_2_^●−^ is the primary ROS produced from a variety of sources [4], while H_2_O_2_ is not a radical but can generate OH^●^ via the Fenton reaction in the presence of Fe^2+^, which is one of the most harmful radicals in the human body [5]. Every cell possesses a mechanism that includes both non-enzymatic and enzymatic antioxidants that are able to neutralize excess ROS and protect against the harmful effects of oxidative stress [6]. The main enzymatic protectors are superoxide dismutase (SOD), catalase (CAT), and glutathione peroxidases (GPx). SOD is a metalloprotein that scavenges O_2_^●−^, and converts it into H_2_O_2_ and molecular oxygen O_2_ [7], while H_2_O_2_ is reduced to water by the CAT [8] and GPx [9]. By using those enzymes, not only are two toxic ROS, O_2_^●−^ and H_2_O_2_, are converted into harmless water, but also they prevent the formation of OH^●^ [10]. CAT is responsible for the detoxification of high exogenous and endogenous H_2_O_2_ concentrations, whereas GPx is involved in the removal of low endogenous H_2_O_2_ concentrations. Additionally, the activity of GPx is regulated by the selenium (Se) and glutathione (GSH) concentration, as well as by glutathione reductase (GR) activity [11]. Se, the crucial element for GPx activity, also plays a pivotal role in diverse functions of the CNS and its relevance in the context of autoimmune diseases [12]. An experimental study has shown that the genetic inactivation of selenoprotein P or its neuronal receptor lead to low Se levels in the brain, resulting in spontaneous neurological deficits and neurodegeneration [13]. The balance between the main antioxidant enzymes, including SOD, GPx, and CAT, is believed to be more crucial than the activity of the single enzymes. For example, the ratio of SOD/CAT or even SOD/CAT + GPx has been proved to have better antioxidant ability than either single enzyme [14,15].

In our earlier conducted study, we observed that the concentration of advanced oxidation protein products (AOPP) was significantly higher in patients with relapsing–remitting multiple sclerosis (RRMS) compared to controls (confirmed with ROC analysis). Relationships were also shown between the concentration of AOPP, and markers of inflammation or types of disease-modifying treatment (DMT) used in MS subjects. Obradovic et al. [16] conducted a study on 118 multiple sclerosis patients and found that not only a decreased concentration of AOPP but also increased activity of SOD was associated with better clinical recovery following corticosteroid relapse treatment. Therefore, in the present study, we aimed to evaluate the involvement of the selected antioxidant enzymes including SOD, GPx, and CAT as well as other oxidative stress parameters such as total oxidant (TOS) and antioxidant status (TAS) and lipid hydroperoxide (LPO) in the blood of RRMS patients. The concentration of interleukin-6 (IL-6) was used as an inflammatory marker. We also evaluated the relationships between changes in pro/antioxidant balance and demographics and MS-related clinical issues.

## 2. Materials and Methods

The study comprised patients diagnosed with RRMS, either admitted to the hospital or attending consultations at the Department of Neurology, Wroclaw Medical University, during the period from May to July 2021. The diagnosis of RRMS was confirmed using McDonald’s criteria for MS [17]. All patients in the study were receiving DMT and attending regular follow-up appointments. Their medical records offered comprehensive documentation of the progression of their disease.

Participants were excluded if they had the primary or secondary progressive type of MS, had experienced an MS relapse within the last 3 months, had initiated or changed disease-modifying therapy (DMT) within the last 6 months, had suffered from uncontrolled systemic comorbidities, or had any addictions, such as smoking or alcohol abuse. By implementing these exclusion criteria, the research aimed to focus on a well-defined and homogeneous group, reducing potential confounding factors and enhancing the validity of the findings (13). Finally, 161 patients were included (67 men and 94 women, aged 24–69 years, median 43.0).

On the basis of medical records, duration of disease, degree of disability (expressed as Expanded Disability Status Scale (EDSS) score) [18,19] and type of DMT were determined. DMT used in the study group included: glatiramer acetate (GA) (n = 20), interferon beta (IFNβ-1a, IFNβ-1b, pegylated interferon beta) (n = 43), teriflunomide (TER) (n = 29), fingolimod (FTY) (n = 29), dimethyl fumarate (DMF) (n = 41).

The control group consisted of age-matched 29 healthy individuals (9 men and 20 women, aged 25–68 years, median 41.0).

Venous blood samples were collected into serum and plasma BD Vacutainer tubes (BD Diagnostics, Plymouth, UK) from all patients and healthy subjects. Blood samples were centrifuged at 2500× *g* for 15 min at room temperature to separate plasma and serum. Serum and plasma were frozen at −80 °C until used. Plasma samples were used to measure concentrations of TAS, LPO, nitrate/nitrite (NO_3_/NO_2_), and activity of SOD and GPx, while serum was used to determine the concentration of TOS, interleukin-6 (IL-6), and the activity of CAT.

The study was approved by the Ethics Committee of Wroclaw Medical University, Poland (KBN No 146/2022) and conducted in accordance with the Helsinki Declaration.

Interleukin-6 was measured using the Human IL-6 DuoSet ELISA kit (ref. No: DY206-05; R&D Systems, Minneapolis, MN, USA) and the DuoSet ELISA Ancillary Reagent Kit 2 (ref. No: DY008B, R&D Systems, Minneapolis, MN, USA). Firstly, the microplate wells were coated with specific antibodies against human IL-6. After coating, a blocking step was performed to prevent non-specific binding. Either samples or standards were added to the wells. Then detection antibodies specific to IL-6 and streptavidin-horseradish peroxidase were pipetted into each well. In the next step a substrate solution, which undergoes a colorimetric reaction in the presence of horseradish peroxidase, was added to each well. Finally, a stop solution was pipetted to halt the color development reaction. The absorbance was measured at a wavelength of λ = 450 nm using a spectrophotometer.

Total oxidant status (TOS) was measured in serum using the earlier described method [20], in which the oxidants present in the sample oxidize the ferrous ion-o-dianisidine complex to ferric ion (o-dianisidine dihydrochloride, ref. No: D3235; ammonium iron (II) sulfate hexahydrate, ref. No: 203504; both reagents from Sigma-Aldrich, Taufkirchen, Germany). Ferric ion produces a colored complex with xylenol orange (ref. No: 398187, Sigma-Aldrich, Taufkirchen, Germany). The color intensity, which can be measured spectrophotometrically at λ = 340 nm, is related to the total amount of oxidant molecules present in the sample.

The concentration of Nitrate/Nitrite (NO_3_/NO_2_) was measured using a total nitric oxide and nitrate/nitrite parameter assay kit (ref. No: KGE001, R&D Systems, Minneapolis, MN, USA). This assay quantifies nitric oxide concentrations using the enzymatic conversion of nitrate to nitrite, facilitated by nitrate reductase. The subsequent reaction involves the colorimetric detection of nitrite using the Griess reaction, which results in the formation of an azo dye product. The absorption of light by this azo-derivative occurs at a wavelength of λ = 570 nm.

Lipid peroxidation parameters were measured in plasma using a lipid peroxidation assay kit (ref. No: KB03002, BQC Redox Company, Asturias, Spain). The assay kit is likely designed to measure the concentration of malondialdehyde (MDA) and 4-hydroxynonenal (HNE), which are products of lipid peroxidation during the propagation phase. The absorbance of the MDA and HNE products was measured spectrophotometrically at a specific wavelength of λ = 586 nm.

Total antioxidant capacity was measured using an antioxidant capacity test (ref. No: IC5200, Immuchrom, Heppenheim, Germany) through a process involving the reaction of antioxidants present in the serum with a specific quantity of added H_2_O_2_. The antioxidants in the serum neutralize a portion of the introduced H_2_O_2_, and the remaining H_2_O_2_ is then measured using a spectrophotometric method. This involves an enzymatic reaction that converts TMB (3,3′,5,5′-tetramethylbenzidine) into a colored product, allowing for the quantification of the residual H_2_O_2_ and, consequently, the antioxidative capacity of the serum. After addition of a stop solution the samples were measured at λ = 450 nm. The quantification was performed by the delivered calibrator (ref. No: IC5200ka, Immuchrom, Heppenheim, Germany).

The oxidative status index (OSI) was also calculated proportioning TOS to TAS concentration.

Cu/Zn SOD activity was determined with a SOD assay kit (ref. No: 19160, Sigma-Aldrich, Taufkirchen, Germany). The measurement of SOD activity involves using a highly water-soluble tetrazolium salt known as WST-1 [2-(4-Iodophenyl)-3-(4-nitrophenyl)-5-(2,4-disulfophenyl)-2H-tetrazolium, monosodium salt]. When this salt is exposed to a superoxide anion, it undergoes reduction, producing a water-soluble formazan dye. The rate of this reduction, influenced by xanthine oxidase activity, exhibits a linear relationship with the concentration of superoxide anions. However, the presence of SOD inhibits this reduction process. To quantify the activity of SOD, the 50% inhibition activity (IC50) is determined using a colorimetric method. The level of superoxide anions is proportional to the absorbance at λ = 440 nm, and the inhibition activity of SOD can be assessed by measuring the decrease in the color development. By employing this approach, the SOD activity can be accurately quantified, providing valuable insights into the antioxidative capacity of the sample under investigation.

The assay of GPx activity was performed with a glutathione peroxide assay kit (ref No: MAK437, Sigma-Aldrich, Taufkirchen, Germany). The activity of GPx was measured indirectly by a coupled reaction with GR. Oxidized glutathione, formed during the reduction of hydroperoxide by GPx, is regenerated back to its reduced state through the actions of GR and nicotinamide adenine dinucleotide phosphate (NADPH). The method directly measures NADPH consumption in the enzyme coupled reactions. The decrease in the absorbance at λ = 340 nm is directly proportional to the enzyme activity in the sample.

The activity of CAT was assayed using a catalase assay kit (ref. No: MAK381, Sigma-Aldrich, Taufkirchen, Germany). In the assay, catalase first reacts with H_2_O_2_ to produce H_2_O and O_2_. The unconverted H_2_O_2_ in the assay subsequently reacts with a specific probe (in dimethylsulfoxide (DMSO)) present in the assay kit to produce a chromogenic product. The absorbance of the chromogenic product was measured spectrophotometrically at a wavelength of λ = 570 nm. Catalase activity was inversely proportional to the absorbance.

### Statistics

Values were expressed as mean and standard deviation (X ± SD) as well as median and 1st quartile and 3rd quartile. Normality of variables was tested using the Shapiro–Wilk test. Homogeneity of variance was assessed using Levene’s test. In case of a lack of normal distribution and variance uniformity, the differences between two groups (studied vs. control subgroups and male vs. female subgroups) were investigated using a non-parametric U Mann–Whitney test, while among five subgroups treated with specific DMT using Kruskal–Wallis one-way analysis of variance by ranks. Furthermore, post hoc analysis was performed to show a specific difference between groups.

For all individual parameters we calculated the receiver operating characteristic (ROC) curve analysis. Furthermore, Youden’s index is calculated using the following formula: Youden’s index = Sensitivity + Specificity − 1 to assess the ability of investigated parameters to discriminate between patients with RRMS (positive group) and the control group (negative group).

The correlation was determined using Spearman’s rank-order correlation coefficient. In all instances, *p* < 0.05 was considered statistically significant. Statistical analysis was performed using the Statistica Software Package, version 13.3 (Polish version; StatSoft, Kraków, Poland).

## 3. Results

The specific characteristics and investigated parameters in the blood of the patients with RRMS and the control group are summarized in Table 1. The mean age did not differ between the two groups. However, there were notable differences in certain parameters between the RRMS group and the control group. Patients with RRMS had a significantly higher concentration of IL-6, NO_2_ and activity of CAT, while the activity of GPx was decreased when compared to the control group. Furthermore, the calculated value of SOD/GPx + CAT was almost 2-fold higher in the group of patients with RRMS compared to the control group. The values of other parameters were found to be similar in both groups.

The ROC curve for all investigated parameters was plotted in the diagnosis of RRMS as shown Figure 1a–c and Table 2. GPx with AUC = 0.831 (x = 0.3387; y = 0.92) has the highest values as single parameter form whole analyzed parameters for distinguishing patients with RRMS from control group. While OSI with AUC = 0.533 (x = 0.5378; y = 0.7639); CAT with AUC = 0.312 (x = 0.9572; y = 1.013) and IL-6 with AUC = 0.351 (x = 0.9933; y = 1) is not sufficient to distinguish patients with RRMS from the control group. Furthermore, we did not confirm the usefulness of determining the concentrations of TOS, NO_2_, NO_3_, LPO, TAS, nor the activity of SOD and the value of the SOD/GPx + CAT ratio to differentiate RRMS from controls. We were not able to determine the value to discriminate patients with RRMS and the control group for the concentration of NO_2_, NO_3,_ or the value of the SOD/GPx + CAT ratio (Figure 1b,c).

In the group of patients suffering from RRMS we performed the pairwise correlation. We did not find any significant correlation between age, disease duration and EDSS, and all analyzed parameters. A significant negative correlation between SOD and GPx activity as well as between GPx and CAT activity was revealed. The value of the SOD/GPx + CAT ratio was positively correlated with the activity of SOD, while negatively with the activity of GPx and CAT. The highest value of coefficient correlation was found for SOD/GPx + CAT and GPx (r = −0.95, *p* < 0.0000). The concentration of AOPP (earlier published data) was negatively associated with GPx activity and positively with CAT activity (Table 3).

When studying both groups (patients with RRSM and the control group) divided by gender, significant differences were observed between male and female patients with RRMS. In the male RRMS patients there was a significantly higher concentration of TAS, and decreased activity of CAT compared to their female counterparts. No significant differences were observed between male and female participants in the control group (Table 4).

All investigated variables were compared among the subgroups of patients treated with DMT. There were statistically significant differences between these subgroups in the disease duration and EDSS score. Differences were also observed in the concentration of IL-6, TOS, LPO, TAS, as well as in the activity of CAT, while the activity of SOD and GPx as well as the value of SOD/GPx + CAT was similar across the subgroups (Table 5).

We found significant changes in the concentrations IL-6, TOS, TAS and the value of OSI between patients treated with TER and DMF. Additionally, in the patients treated with TER also the concentration of TAS was increased when compared to the patients treated with INFs. Furthermore, the concentrations of IL-6 and LPO differed between patients treated with FTY and GA or DMF. A significantly higher concentration of IL-6 was also found in the patients treated with GA rather than TER. In case of antioxidant enzymes, we only observed significant differences in the activity of CAT between patients treated with GA and DMF (Figure 2a–d).

## 4. Discussion

Recent studies have demonstrated that oxidative stress is a prominent feature of the background of multiple sclerosis [2,19,22]. Hence, examining the changes in the pro/oxidant balance in the blood of RRMS patients may provide evidence for diagnostic and therapeutic implications. Additionally, it has been well-documented that the expression patterns of antioxidant enzymes and their protective responses during oxidative injury, neurodegeneration, and MS are varied [23]. Therefore, in the present study, we not only examined the concentration of selected parameters associated with pro/antioxidant balance in the blood of patients with RRMS but also assessed their relationships with clinical MS-related factors.

This is the second part of our investigation, following our previous paper [21] where we demonstrated the utility of determining AOPP concentration to differentiate patients with RRMS from the control group. The prognostic value of AOPP and activity of SOD in foreseeing recovery from MS relapse following corticosteroid treatment was earlier demonstrated by Obradovic et al. [16]. However, studies investigating SOD activity in the serum of MS patients by other authors have yielded mixed results, which is what inspired us to measure both these parameters.

In our study, we did not observe any significant differences in SOD activity between patients with RRMS and the controls. Also, the value of Youden’s index = 0.01 confirmed that the activity of SOD is not a specific parameter for RRMS patients treated with glatiramer acetate, interferon beta (IFNβ-1a, IFNβ-1b, pegylated interferon beta), teriflunomide, fingolimod, or dimethyl fumarate. The similarity in SOD activity between patients with RRMS and the healthy group could be attributed to effective DMT. Other studies have demonstrated changes in SOD gene expression and activity during acute MS attacks [24]. A study conducted by Ljubisavljevic et al. [25] revealed higher SOD activity in patients with clinically isolated syndrome (CIS) and RRMS compared to the control group. However, the SOD induction was more pronounced in CIS patients, which could be a result of direct ROS effects. In RRMS, SOD showed a decrease in activity compared to the CIS group, possibly due to irreversible inactivation caused by prolonged oxidative stress and its by-products. Additionally, SOD activity was found to be lower in both RRMS and CIS patients with higher EDSS scores and a greater number of total radiological lesions in MRI brain scans [25], confirming that changes in SOD activity were observed in more severe forms of MS. Similarly, the significant relationship between SOD activity and EDSS was observed in SPMS patients, which is a more severe type of MS than RRMS [26]. However, the present study included only patients with RRMS and an EDSS score ranging from 0 to 4.5, which could explain the lack of correlation between SOD activity and EDSS. Interestingly, when we analyzed the difference in SOD activity between patients with RRMS and EDSS > 2.0 (67.5 (63.5–70.1)) and ≤ 2.0 (64.7 (61.4–66.4)), we found higher SOD activity in the group with a greater degree of disability (data not published). This finding further suggests that significant changes in SOD activity could be associated with the severity of MS.

Despite not observing significant changes in the activity of SOD, we found decreased activity of GPx and increased activity of CAT in the blood of patients with RRMS compared to the control group. Our findings align with the results reported by Jansen et al. [27], who observed approximately 35–50% decrease in GPx activity in the lymphocytes and granulocytes of MS patients. In our study, plasma GPx activity was 32% lower than in the control group. It is possible that individuals with reduced GPx activity are more susceptible to oxidative damage to membrane fatty acids and functional proteins, leading to neurotoxic damage [28], because GPx not only reduces H_2_O_2_ to water but also prevents lipid hydroperoxides. Another important factor could be associated with genetic and epigenetic predisposition, which could explain the higher GPx activity observed in the serum of Mexican patients with RRMS compared to healthy controls [29]. Nevertheless, in our study we did not observe a significant correlation between LPO concentration and GPx activity. Furthermore, the inflammatory environment in MS, characterized by elevated pro-inflammatory cytokines and oxidative stress, may also contribute to the inhibition of GPx activity [19] but also in this case we did not find significant correlation between GPx activity and IL-6 concentration.

When we analyzed the relationships between SOD, GPx, and CAT activity we showed the negative correlation between SOD activity and GPx activity as well as between GPx activity and CAT activity. It is possible that a negative correlation between SOD and decreased GPx activity is associated with a higher concentration of H_2_O_2_, which can explain higher activity of CAT, because this enzyme catalyzes the disproportionation of H_2_O_2_ to water and oxygen, when a high concentration of H_2_O_2_ occurs. Furthermore, it was reflected by a negative correlation between GPx and CAT activity. ROC analysis (AUC = 0.831; *p* = 0.0000) potentially indicates the clinical usefulness of GPx activity and its diagnostic power to differentiate patients with RRMS from healthy subjects.

Generally, research on the activity of CAT in the peripheral blood of patients with MS has yielded inconclusive results. Some studies have reported decreased CAT activity in the granulocyte lysates of MS patients, while others have shown increased CAT activity in the cerebrospinal fluid and serum of patients with RRMS and CIS compared to the control group. In our present study, we observed an increase in CAT activity in the plasma of patients with RRMS. This could be a compensatory response to counteract oxidative stress when GPx activity is depressed, and it may serve as an adaptive response to the elevated levels of ROS [30]. It is possible that due to a higher concentration of H_2_O_2_, potentially associated with depressed GPx activity, CAT plays a major role in neutralizing this ROS. The activity of CAT was also influenced by gender and DMT. Earlier reports have also suggested that changes in CAT activity could be related to the pathogenesis of many age-associated degenerative diseases [31] as well with human illnesses associated with oxidative stress including inflammatory diseases [32]. It is worth noting that H_2_O_2_ has a longer half-life compared to O_2_^●−^, and unlike superoxide, H_2_O_2_ can easily traverse lipid membranes or be transported through channels [9]. This characteristic may help explain the notable alterations in GPx and CAT activity, while no significant changes are observed in the case of SOD.

In line with the suggestion made by de Haan et al. [15] that the SOD/GPx + CAT ratio exhibits better antioxidant ability than either single enzyme alone, we also included this ratio in our study. A significantly higher value of the SOD/GPx + CAT ratio was observed in the group of patients with RRMS compared to the control group. An experimental study conducted on mice has demonstrated that an altered SOD/GPx + CAT ratio was found in the brain of aging mice and was correlated with increased lipid damage [15]. Therefore, it can be presumed that changes in the value of this ratio could be associated with CNS disorders. In our study, none of the clinical factors affected the value of the SOD/GPx + CAT ratio, also we did not detect a significant correlation between LPO concentration and the value of the SOD/GPx + CAT ratio. Generally, the increased value of the SOD/GPx + CAT ratio was associated with decreased GPx activity (r = −0.95; *p* = 0.000). It seems that various changes in the investigated enzymes (decreased activity of GPx, increased activity of CAT, and unchanged activity of SOD) indicate differential regulation of these antioxidant enzymes, potentially involving transcriptional regulation and post-translational modifications. It is also possible that the expression of CAT is more strongly induced or regulated in patients with RRMS compared to GPx and SOD. In more advanced stages of the disease or during acute exacerbations, there may be a higher demand for antioxidant enzymes to counteract increased oxidative stress, resulting in significant changes in the SOD/GPx + CAT ratio. Unfortunately, as mentioned above, we did not observe any significant correlations between the value of this ratio and age, disease duration, EDSS score, and DMT.

Like the activity of SOD, the ROC analysis also indicated that the activity of CAT and the value of the SOD/GPx + CAT ratio are not a specific parameter for patients with RRMS. A Youden’s index value of 0.05 (AUC = 0.312; *p* = 0.0024) for CAT activity was insufficient to distinguish patients with RRMS from the control group, especially considering that this enzyme is involved in various physiological processes.

In our study, we also analyzed oxidative stress parameters TOS, NO_2_, NO_3_ and LPO. A systematic review and meta-analysis conducted by Zhang et al. [33] in 2020 summarized oxidative stress parameters in the cerebrospinal fluid and blood of patients with MS. The review included 31 studies with 2001 MS patients and 2212 healthy individuals, and it revealed an increased concentration of lipid peroxidation in the blood of patients with MS. However, the levels of TOS and TAS, as well as the activity of SOD, did not differ significantly between patients with MS and the control group [33].

In our study we also did not find any significant changes in the concentration of TOS, LPO, NO_3_, and TAS values in the plasma of patients with RRMS compared to the control group. Similar to our earlier conducted study [21], neither the ferric reducing ability of plasma (FRAP) nor TAS values are parameters able to distinguish the antioxidant capacity between the RRMS and control group. Additionally, these parameters did not correlate with assayed enzyme activity. However, the concentrations of TOS, LPO, and TAS were significantly different among subgroups of patients treated with specific DMT, which confirmed that disorders in pro/antioxidant balance are associated not only with the disease but also with the type of treatment during RRMS.

Meanwhile, the concentration of NO_2_ was increased in the blood of patients with RRMS in comparison to the control group. Nitric oxide (NO) is a free radical gas produced naturally in the body and plays various roles in physiological processes. One of its important derivatives is NO_2_, which is a reactive nitrogen species and a form of oxidized nitrogen. When the concentration of NO_2_ is increased in the blood, it can lead to a condition known as nitrate stress. In the context of RRMS, oxidative stress can further contribute to the inflammation and damage of the CNS, worsening the disease progression [34]. Nevertheless, also in this case we did not observe a significant correlation between other investigated parameters. To establish a conclusive relationship between NO_2_ and nitrate stress in RRMS, further research and investigation are required.

Generally, the assessment of antioxidant capacity can be performed in two ways: by measuring the concentrations of individual antioxidants (such as SOD, GPx, CAT) or by measuring the total antioxidant capacity (TAC, TAS, or FRAP), which provides an estimation of the overall antioxidant components in a sample. While the latter approach is more convenient, requiring less effort, time, and cost, our results clearly shown that the determination of individual antioxidants is more useful for evaluating changes in the pro/antioxidant balance in the blood of patients with RRMS. This approach allows for a more detailed understanding of the antioxidant system and its potential dysregulation in RRMS. While total antioxidant capacity provides a global assessment, it may not capture specific changes in individual antioxidants or their interactions. Therefore, after determination of the concentration of both FRAP and TAS, we advocate for the evaluation of various antioxidants individually to better elucidate the pro/antioxidant balance and its implications in the context of RRMS.

There were several limitations to the current study. The investigation was performed in the blood not in CSF, which is one of the diagnostic hallmarks of MS. We acknowledge that including CSF data could potentially add further depth and confirmation of some of our findings. In future research, we would certainly consider expanding our investigation to incorporate CSF analysis, especially if the resources and the study design permit such an approach. Secondly, due to the small size of the control group our study should be treated as preliminary. Furthermore, it should be noted that there were differences in the number of patients, disease duration, and EDSS scores among the subgroups that received DMT. Therefore, the findings should be interpreted cautiously. Additionally, it is important to note that GPx is involved in various processes in the human body. For future studies, we plan to investigate the activity of GR, glutathione S-transferase, as well as the concentrations of reduced and oxidized GSH, Se, and NADPH. Moreover, the investigation of genetic backgrounds is important in evaluating changes in the pro/antioxidant balance. This will allow us to further explore the potential causes of the decreased activity of GPx during RRMS.

## 5. Conclusions

In summary, the assessment of pro/antioxidant balance parameters indicates their significant alterations in RRMS patients. Specifically, there is a significant decrease in GPx activity in patients with RRMS and increased CAT activity. We propose that the activity of GPx could be a promising marker and/or potential therapeutic target in RRMS. Further investigation is warranted to explore the potential factors contributing to the drop in GPx activity.

## Figures and Tables

**Figure 1 antioxidants-12-01638-f001:**
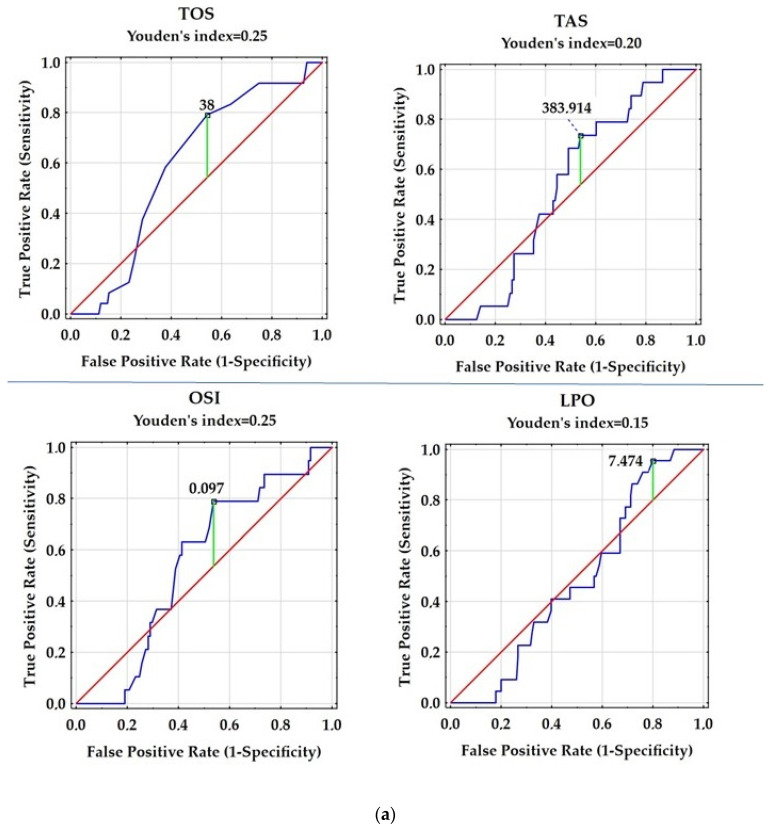
(**a**) Receiver operating characteristic (ROC) curves evaluating the ability of TOS, TAS, LPO concentration or the value of the OSI ratio to distinguish patients with RRMS and control group. (**b**) Receiver operating characteristic (ROC) curves evaluating the ability of NO_2_, NO_3_ or IL-6 concentration to distinguish patients with RRMS and control group. (**c**) Receiver operating characteristic (ROC) curves evaluating the ability of SOD, GPx, CAT activity or the value of the SOD/GPx + CAT ratio to distinguish patients with RRMS and control group.

**Figure 2 antioxidants-12-01638-f002:**
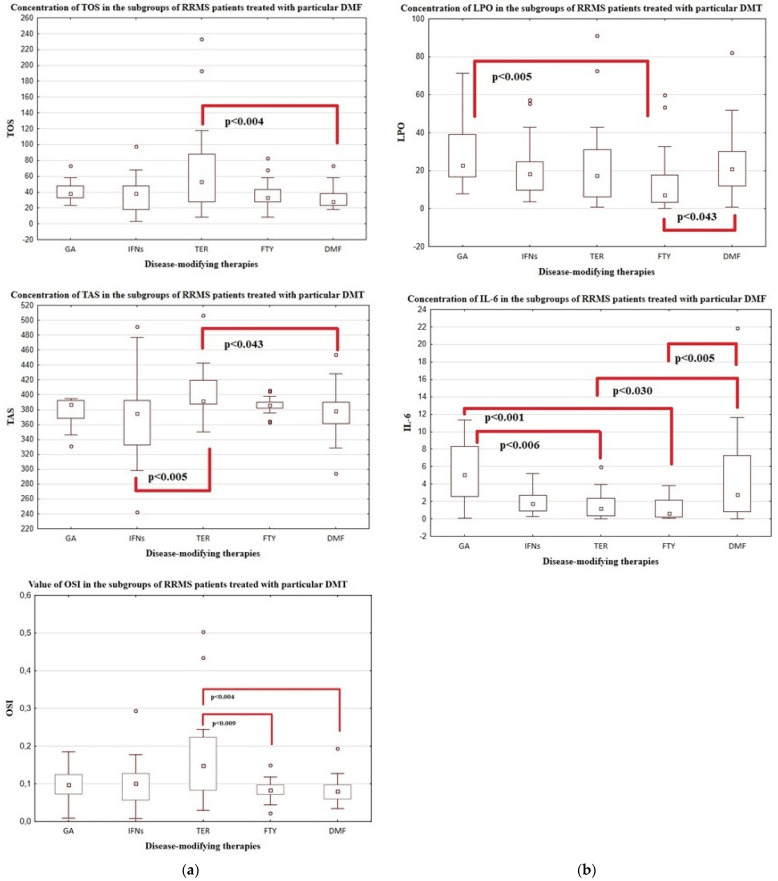
(**a**,**b**) The concentration of TOS, TAS and the value of OSI (**a**) and LPO and IL-6 (**b**) in the blood of patients with RRMS divided according to particular DMT with statistical analysis. (**c**) The concentration of NO_2_ and NO_3_ in the blood of patients with RRMS divided according to particular DMT with statistical analysis. (**d**) The activity of SOD, GPx, CAT and the value of the SOD/GPx + CAT ratio in the blood of patients with RRMS divided according to particular DMT with statistical analysis. Legend: □—median value; ᴏ—outliers.

**Table 1 antioxidants-12-01638-t001:** The specific characteristics and investigated parameters in patients with RRSM and control group.

Variable	Control Group	Patients with RRMS
Number of subjects	n = 29	n = 158
Sex [men/women]	9/20	65/93
Age [years]	41.9 ± 10.1 41.0 (36.0–41.0)	43.9 ± 9.7 43.0 (37.0–51.0)
Duration of RRMS [years]	N/A	12.3 ± 6.4 12.0 (8.0–15.0)
EDSS	N/A	2.8 ± 1.5 2.2 (1.5–3.5)
IL-6 [pg/mL]	1.6 ± 2.0 0.9 (0.4–2.7)	4.4 ± 7.1 2.1 (0.7–5.9) *
TOS [µM]	44.0 ± 17.7 43.0 (38.0–48.0)	58.4 ± 96.5 38.0 (23.0–53.0)
NO_2_ [µmol/L]	6.6 ± 2.4 6.7 (5.9–7.6)	11.6 ± 4.6 11.4 (7.43–15.5) *
NO_3_ [µmol/L]	10.2 ± 8.0 8.3 (4.5–15.8)	18.3 ± 28.5 10.4 (5.5–14.6)
LPO [µM]	21.9 ± 19.0 16.4 (11.4–24.3)	28.0 ± 40.1 18.7 (8.0–30.1)
TAS [µM]	370.7 ± 61.5 387.9 (383.5–390.4)	380.3 ± 43.9 385.5 (368.5–392.8)
OSI	0.1 ± 0.0 0.11 (0.08–0.13)	0.2 ± 0.3 0.10 (0.07–0.14)
SOD [U/L]	65.6 ± 10.7 64.0 (59.8–69.5)	66.6 ± 8.5 65.8 (61.9–71.5)
GPx [U/L]	113.6 ± 71.1 98.2 (86.4–113.9)	69.6 ± 28.4 66.8 (51.1–82.5) *
CAT [U/L]	4.9 ± 1.4 5.1 (3.6–5.8)	5.8 ± 1.8 6.0 (5.0–6.7) *
SOD/GPx + CAT	0.6 ± 0.2 0.6 (0.6–0.7)	1.2 ± 1.5 1.0 (0.7–1.2) *

Values were shown as X ± SD and median with 1st quartile, and 3rd quartile; * *p* < 0.05 when compared to a control group. RRMS—relapsing–remitting multiple sclerosis; EDSS—expanded disability status scale; IL-6—interleukin-6; TOS—total oxidant status; NO_2_—nitrite; NO_3_—nitrate; LPO—lipid peroxide; TAS—total antioxidant status; OSI—oxidative status index; SOD—superoxide dismutase; GPx—glutathione peroxidase; CAT—catalase; N/A—not applicable.

**Table 2 antioxidants-12-01638-t002:** The diagnostic efficiency of single biomarkers to distinguish RRMS patients from healthy persons.

Variable	AUC	SE	AUC Lower 95%	AUC Upper 95%	z = (v1 − 0.5)/v2	*p* Value
IL-6 [pg/mL]	0.351	0.067	0.221	0.482	−2.233	0.0255
TOS [µM]	0.58	0.053	0.476	0.683	1.513	0.1304
NO_2_ [µmol/L]	0.004	0.004	−0.003	0.011	−136.109	0.000
NO_3_ [µmol/L]	0.211	0.057	0.1	0323	−5.058	0.000
LPO [µM]	0.483	0.054	0.377	0.589	−0.311	0.7555
TAS [µM]	0.536	0.055	0.429	0.643	0.654	0.5129
OSI	0.533	0.059	0.417	0.649	0.551	0.5816
SOD [U/L]	0.415	0.063	0.291	0.538	−1.359	0.1740
GPx [U/L]	0.831	0.044	0.745	0.916	7.586	0.0000
CAT [U/L]	0.312	0.062	0.191	0.434	−3.036	0.0024
SOD/GPx + CAT	0.214	0.049	0.118	0.31	−5.856	0.0000

AUC—area under the curve; IL-6—interleukin 6; TOS—total oxidant status; NO_2_—nitrite; NO_3_—nitrate; LPO—lipid peroxide; TAS—total antioxidant status; OSI—oxidative status index; SOD—superoxide dismutase; GPx—glutathione peroxidase; CAT—catalase.

**Table 3 antioxidants-12-01638-t003:** Correlation coefficients among analyzed parameters in the group of patients with RRMS.

Correlation	SOD	GPx	CAT
IL-6 [pg/mL]	NS	NS	NS
TAS [µM]	NS	NS	NS
TOS [µM]	NS	NS	NS
NO_2_ [µmol/L]	NS	NS	NS
NO_3_ [µmol/L]	NS	NS	NS
OSI	NS	NS	NS
SOD [U/L]	-	−0.27; 0.039	NS
GPx [U/L]	−0.27; 0.039	-	−0.37; 0.005
CAT [U/L]	NS	−0.37; 0.005	-
SOD/GPx + CAT	0.46; 0.000	−0.95; 0.000	0.29; 0.027
AOPP [µM] #	NS	−0.30; 0.024	0.26; 0.039

IL-6—interleukin; TAS—total antioxidant status; TOS—total oxidant status; NO_2_—nitrite; NO_3_—nitrate; SOD—superoxide dismutase; GPx—glutathione peroxidase; CAT—catalase; NS—not significant. #—earlier published result [21].

**Table 4 antioxidants-12-01638-t004:** The specific characteristics and investigated parameters in patients with RRSM and the control group divided according to the gender.

Variable	Control Group		Patients with RRMS	
	Women	Men	Women	Men
Age	42.1 ± 11.2 41.0 (33.0–49.0)	41 ± 7.5 41.0 (36.0–47.5)	42.7 ± 8.5 43.0 (37.0–51.0)	43.8 ± 9.4 42.5 (35.5–51.0)
Disease duration [years]	N/A	N/A	11.9 ± 5.7 12.0 (8.0–15.0)	13.0 ± 6.7 12.0 (8.5–16.0)
EDSS	N/A	N/A	2.8 ± 1.4 2.5 (1.5–3.5)	2.7 ± 1.6 2.5 (1.5–3.0)
IL-6 [pg/mL]	1.4 ± 2.0 0.8 (0.2–1.4)	2.1 ± 2.4 1.0 (0.3–2.7)	4.3 ± 6.7 2.2 (0.8–5.2)	4.6 ± 7.9 1.5 (0.6–5.9)
TOS [µM]	39.4 ± 12.9 39.4 (33.0–48.0)	48.0 ± 14.4 48.0 (40.5–50.5)	55.9 ± 102.0 38.0 (23.0–48.0)	62.6 ± 87.1 38.0 (25.5–58.0)
NO_2_ [µmol/L]	6.7 ± 2.5 6.7 (5.9–48.0)	6.3 ± 2.0 6.8 (6.3–7.4)	21.9 ± 8.4 21.0 (13.8–27.2)	26.0 ± 10.2 26.2(15.5–34.1)
NO_3_ [µmol/L]	10.9 ± 8.4 9.6 (4.6–17.3)	6.9 ± 5.5 6.0 (3.3–8.1)	24.2 ± 19.8 20.1 (11.9–27.9)	70.6 ± 100.6 20.8 (10.1–150.1)
LPO [µM]	18.8 ± 9.6 15.9 (12.2–28.0)	17.5 ± 6.6 17.5 (9.8–24.1)	24.4 ± 36.4 17.2 (8.0–24.6)	33.6 ± 44.9 20.6 (9.1–37.7)
TAS [µM]	381.7 ± 14.3 386.1 (371.3–392.0)	387.8 ± 4.4 388.4 (387.7–390.2)	374.5 ± 5 382.3 (361.5–391.3)	390.4 ± 41.4 390.2 (378.0–394.2) *
OSI	0.1 ± 0.0 0.1 (0.1–0.1)	0.1 ± 0.0 0.1 (0.1–0.1)	0.2 ± 0.3 0.1 (0.1–0.1)	0.2 ± 0.4 0.1 (0.1–0.2)
SOD [U/L]	64.9 ± 9.0 64.0 (59.8–71.1)	62.4 ± 5.7 63.0 (59.1–66.0)	66.8 ± 7.3 66.2 (62.2–71.8)	66.3 ± 10.3 64.6 (60.8–70.9)
GPx [U/L]	100.6 ± 26.0 98.2 (78.6–113.9)	148.7 ± 123.9 100.2 (96.3–129.4)	68.2 ± 30.6 62.9 (51.1–86.4)	72.2 ± 24.1 74.6 (55.0–78.6)
CAT [U/L]	4.7 ± 1.5 4.4 (3.2–5.7)	5.4 ± 1.0 5.5 (5.0–5.9)	5.8 ± 1.7 6.0 (5.1–6.7)	5.8 ± 4.9–6.3 5.8 (4.9–6.3) *
SOD/GPx + CAT	0.7 ± 0.2 0.6 (0.6–0.8)	0.5 ± 0.2 0.6 (0.5–0.6)	1.4 ± 1.8 1.0 (0.7–1.2)	1.0 ± 0.4 0.8 (0.7–1.2)

Values were shown as X ± SD and median with 1st quartile, and 3rd quartile. * *p* < 0.01 when compared to female patients with RRMS. RRMS—relapsing–remitting multiple sclerosis; EDSS—expanded disability status scale; TOS—total oxidant status; NO_2_—nitrite; NO_3_—nitrate; LPO—lipid peroxide; TAS—total antioxidant status; OSI—oxidative status index; SOD—superoxide dismutase; GPx—glutathione peroxidase; CAT—catalase; N/A—not applicable.

**Table 5 antioxidants-12-01638-t005:** Comparison of investigated variables in the subgroups of RRMS patients treated with particular DMT.

Variable	Patients with RRMS				
	GA	IFNs	TER	FTY	DMF
Number of subjects	n = 18	n = 47	n = 26	n = 28	n = 39
men/women [number]	6/12	23/24	14/12	8/20	14/25
Age [years]	43.2 ± 10.2 44.0 (36.0–51.0)	44.9 ± 8.6 43.0 (39.0–50.0)	47.6 ± 12.3 46.5 (42.0–58.0)	42.8 ± 10.4 41.0 (34.5–49.5)	42.6 ± 10.4 42.0 (35.0–50.0)
Disease duration [years]	12.2 ± 6.2 10.0 (7.0–17.0)	13.0 ± 4.9 12.0 (10.0–15.0)	13.1 ± 7.6 13.5 (7.0–17.0)	15.5 ± 5.3 14.0 (12.5–20.5)	10.5 ± 7.7 8.0 (4.0–13.0) *
EDSS	2.4 ± 1.3 2.0 (1.5–3.0)	2.2 ± 1.0 2.0 (1.5–3.0)	3.1 ± 1.4 3.0 (2.0–4.0)	4.3 ± 1.8 4.0 (3.0–6.0)	2.3 ± 1.3 2.0 (1.0–3.0) *
IL-6 [pg/mL]	5.3 ± 3.4 6.7 (2.8–11.3)	3.1 ± 4.0 2.0 (1.0–3.2)	1.5 ± 1.5 1.3 (0.4–2.6)	1.3 ± 1.2 1.3 (0.4–3.4)	3.8 ± 3.4 2.9 (0.8–7.5) *
TOS [µM]	37.7 ± 16.5 38.0 (28.0–48.0)	56.4 ± 91.4 38.0 (28.0–48.0)	126.7 ± 182.7 58.0 (33.0–103.0)	41.8 ± 30.1 35.5 (28.0–43.0)	35.2 ± 21.6 28.0 (23.0–38.0) *
NO_2_ [µmol/L]	30.0 ± 7.4 31.4 (25.5–33.4)	20.7 ± 9.3 16.0 (14.1–28.6)	22.2 ± 9.2 20.5 (16.2–28.1)	22.3 ± 11.7 20.7 (13.1–24.8)	22.2 ± 7.5 24.8 (15.5–26.9)
NO_3_ [µmol/L]	15.3 ± 7.7 14.6 (10.1–7.7)	43.3 ± 72.5 21.3 (9.8–41.3)	12.0 ± 13.5 12.0 (2.4–21.6)	28.2 ± 15.7 23.8 (16.9–29.2)	53.4 ± 66.3 22.8 (13.3–87.4)
LPO [µM]	42.1 ± 58.5 23.3 (16.7–42.2)	26.6 ± 34.6 19.1 (11.2–27.2)	23.4 ± 23.5 17.2 (6.2–31.2)	13.3 ± 16.0 7.2 (3.3–17.7)	36.3 ± 53.1 21.4 (13.3–35.1) *
TAS [µM]	377.7 ± 18.5 386.4 (368.6–392.7)	370.3 ± 48.8 376.7 (333.8–392.5)	398.5 ± 62.6 391.5 (385.5–421.6)	380.3 ± 27.4 385.5 (381.0–390.0)	379.8 ± 40.2 378.7 (361.5–390.9) *
OSI	0.1 ± 0.0 0.1 (0.1–0.1)	0.1 ± 0.2 0.1 (0.1–0.1)	0.2 ± 0.4 0.2 (0.1–0.2)	0.1 ± 0.1 0.1 (0.1–0.1)	0.1 ± 0.0 0.08 (0.1–0.1) *
SOD [U/L]	67.8 ± 8.0 66.4 (62.2–68.6)	66.6 ± 7.8 67.2 (63.0–70.9)	68.6 ± 8.0 66.7 (62.7–76.2)	66.3 ± 10.1 65.2 (57.3–74.2)	65.1 ± 9.0 63.8 (60.6–68.3)
GPx [U/L]	57.0 ± 23.9 51.1 (35.4–78.6)	72.1 ± 32.0 66.8 (51.1–94.3)	87.4 ± 53.3 68.8 (53.0–121.8)	74.1 ± 22.3 70.7 (58.9–94.3)	72.7 ± 17.2 72.9 (60.9–78.6)
CAT [U/L]	6.9 ± 1.4 6.6 (6.0–7.1)	5.9 ± 1.1 6.0 (5.1–6.7)	6.0 ± 2.4 6.2 (4.2–7.8)	5.5 ± 1.6 5.1 (4.3–6.1)	4.2 ± 2.4 4.8 (0.8–6.1) *
SOD/GPx + CAT	1.2 ± 0.5 1.1 (0.8–1.6)	1.6 ± 2.4 1.0 (0.7–1.3)	0.8 ± 0.4 0.8 (0.6–1.1)	0.9 ± 0.5 0.7 (0.6–1.2)	0.9 ± 0.2 0.9 (0.8–1.1)

Values were shown as X ± SD and median with 1st quartile, and 3rd quartile. * *p* < 0.02 when compared the values among five different treatment subgroups. RRMS—relapsing–remitting multiple sclerosis; GA—glatiramer acetate; IFNs—interferons; TER—teriflunomide; FTY—fingolimod; DMF—dimethyl fumarate; EDSS—expanded disability status scale; TOS—total oxidant status; NO_2_—nitrite; NO_3_—nitrate; LPO—lipid peroxide; TAS—total antioxidant status; OSI—oxidative status index; SOD—superoxide dismutase; GPx—glutathione peroxidase; CAT—catalase.

## Data Availability

All relevant data in the current study are available from the corresponding author on request.

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
