# Peer review of "Exploring the Relationship between Antioxidant Enzymes, Oxidative Stress Markers, and Clinical Profile in Relapsing–Remitting Multiple Sclerosis"

_antioxidants, 2023, doi:10.3390/antiox12081638_

Round 1
Reviewer 1 Report
Herein, Bizon et al. examined the concentration of selected parameters associated with pro/antioxidant balance in patients with relapsing-remitting multiple sclerosis though the analysis of 158 diseased and 29 control blood samples. The authors focused also on the impact of disease duration such as EDSS score, DMT and gender. Important evidence is provided such as a significant decrease GPx activity in patients with RRMS and increased CAT activity. Some comment for manuscript’s improvement should be taken into account:
1. Figure 2 is complicated. a-c is difficult to recognize. Please correct or reorganized the figure.
2. Regarding ROC curves, which is the biological meaning of AUC lower 95% and AUC upper 95%; Please explain the significance of Youden’s index.
3. In Tables, I suggest authors present the levels of all markers including the standard deviation.
4. In Table 3, p value is refereed among different treatment subgroups. The authors should explain the group they used for comparison.
5. Except of hydrogen peroxide involvement, can the authors estimate other parameters implicated in the negative correlation between SOD, GPx activity and CAT activity. For example, the genetic background of patients could be analyzed and related genes and pathways should be stressed.
6. I am wondering about NADPH levels and NOX2-derived ROS if they have been measured/could be measured before SOD activity in clinical plasma samples? Did the authors check NOX2 activity?
7. Did the authors identify significant changes in the SOD/CAT+GPx ratio between men and women in RRMS patients?
8. The authors could present some more specific targets for future studies in conclusion.
Minor editing of English language required
Author Response
REVIEWER #1
Herein, Bizon et al. examined the concentration of selected parameters associated with pro/antioxidant balance in patients with relapsing-remitting multiple sclerosis though the analysis of 158 diseased and 29 control blood samples. The authors focused also on the impact of disease duration such as EDSS score, DMT and gender. Important evidence is provided such as a significant decrease GPx activity in patients with RRMS and increased CAT activity. Some comment for manuscript’s improvement should be considered:
- Figure 2 is complicated. a-c is difficult to recognize. Please correct or reorganized the figure. Regarding ROC curves, which is the biological meaning of AUC lower 95% and AUC upper 95%; Please explain the significance of Youden’s index.
We thank the Reviewer for taking the time to read, assess the content of the manuscript and give valuable suggestions.
According to point 1:
We attempted to enhance the quality of Figure 2 and we have rewritten Results section.
Regarding to ROC curves, the AUC is a numerical summary of the ROC curve's performance. It represents the area under the ROC curve and ranges from 0 to 1. A perfect classifier has an AUC of 1, while a random or non-discriminative classifier has an AUC of 0.5. The AUC measures the ability of the classifier to rank positive instances higher than negative instances.
The AUC lower 95% and AUC upper 95% values help to quantify the uncertainty associated with the AUC estimate, considering the variability in the data and sample size. If the AUC lower 95% and AUC upper 95% both exceed 0.5 (the no-discrimination value), it suggests that the classifier's performance is significantly better than random. If the interval includes 0.5, it implies that the classifier's performance is not significantly different from random, indicating that it has limited discriminatory power.
Youden’s index is commonly used in medical research and statistics to assess the ability of a diagnostic test to discriminate between two groups, in our study between patients with RRMS (positive group) and control group (negative group). Youden's index is calculated using the following formula:
Youden’s index = Sensitivity (true positive rate) + Specificity – 1 (false positive rate).
The perfect value of Youden's index (J) is 1, which indicating that there are no false positives or false negatives.
In the section of statistical analysis (verse 188-191) as well as on the Figure 2 we introduce an additional information (True Positive Rate (Sensitivity) and False Positive Rate (1-Specificity)).
- In Tables, I suggest authors present the levels of all markers including the standard deviation.
According to point 2:
In all tables we introduced the mean value and standard deviation (X±SD). Nevertheless, we also left the value of median, 1st quartile, and 3rd quartile, because many statistical analyses was performed using non-parametric tests.
- In Table 3, pvalue is refereed among different treatment subgroups. The authors should explain the group they used for comparison.
According to point 3:
Firstly, in Table 3 we performed Kruskal-Wallis one-way analysis of variance by ranks to generally show the differences between five subgroups of patients. The significant value for p was lower than 0.02. Furthermore, for those cases we performed post hoc analysis to show a specific difference between subgroups, which was presented in the Figure 1 a-c. We introduced an additional information in the section Statistical analysis.
- Except of hydrogen peroxide involvement, can the authors estimate other parameters implicated in the negative correlation between SOD, GPx activity and CAT activity. For example, the genetic background of patients could be analyzed and related genes and pathways should be stressed.
According to point 4:
Thank you very much for valuable suggestion. We are aware that the alterations in enzymes activity could have genetic backgrounds. But in this paper, we aimed to determine selected pro/antioxidant balance parameters in the blood of patients with RRMS to found which of them are significantly changed by disease, gender, DMT, disability score and disease duration. After obtained results we will further search the potentially causes of those changes including genetic background:
“For future studies, we plan to investigate the activity of glutathione reductase, glutathione S-transferase, as well as the concentrations of reduced and oxidized glutathione, selenium, and NADPH. Moreover, the investigation of genetic backgrounds is important in evaluating changes in the pro/antioxidant balance. This will allow us to further explore the potential causes of the decreased activity of GPx during RRMS.”
- I am wondering about NADPH levels and NOX2-derived ROS if they have been measured/could be measured before SOD activity in clinical plasma samples? Did the authors check NOX2 activity?
According to point 5:
Yes, we measured the concentration of NO2 and NO3 and we introduced this data into current paper in methods (verses 127-132), results (all tables and Figure 1b and 2c), discussion section (verses 430-439) and as position 34 in references. Unfortunately, we did not assay the activity of NOX2, but we measured the concentration of eNOS (NOS3). We did not find any significant differences in the concentration of NOS3 between all studied groups as well as any significant correlations between its concentration and clinical or biochemical parameters, therefore we did not introduce this enzyme into the paper.
- Did the authors identify significant changes in the SOD/CAT+GPx ratio between men and women in RRMS patients?
According to point 6:
No, we did not reveal any significant changes in the value of SOD/GPx +CAT ratio between women and men in RRMS patients.
- The authorscould present some more specific targets for future studies in conclusion.
According to point 7:
Thank you very much for your suggestion. At the end of the discussion section, we put the information about limitation of the study as well as we extended more specific targets for future studies (verses 453-474).

Reviewer 2 Report
This paper describes antioxidant enzymes and oxidative stress markers in blood of RRMS patients in comparison to controls.
I have the following comments:
1. The amount of controls (n=29) is not well balanced to the patient group (n=158). That could introduce some problems, especially in the gender subanalysis (Tab. 2). For a reliable study the amount of controls must be increased.
2. There is a problem with the IL-6 values between Tab. 1 and 2. In Tab. 1 IL-6 is in patients significantly higher, but the data in Tab. 2 do not show this accordingly.
3. What about CSF data? Usually CSF is available for MS patients. It would be more convincing for the reader if at least some of the results could be confirmed in CSF which is one of the diagnostic hallmarks of MS.
4. Study limitations have not discussed in appropriate detail.
None.
Author Response
REVIEWER #2
This paper describes antioxidant enzymes and oxidative stress markers in blood of RRMS patients in comparison to controls.
I have the following comments:
- The number of controls (n=29) is not well balanced to the patient group (n=158). That could introduce some problems, especially in the gender subanalysis (Tab. 2). For a reliable study the number of controls must be increased.
According to point 1:
We thank the Reviewer for taking the time to read, assess the content of the manuscript and give valuable suggestions.
We appreciate the Reviewer’s suggestion regarding the imbalance between the number of controls (n=29) and the patient group (n=158) in our study. It is essential to maintain a balanced and representative sample size for a reliable and robust study. Despite our best efforts to recruit more controls, the number of available willing participants was limited within the given timeframe of the study.
We agree that a larger and more balanced control group would indeed strengthen the statistical power and validity of our findings, particularly in subgroup analyses such as the gender subanalysis (Tab. 2). A larger control group helps to reduce the risk of confounding factors and improves the generalizability of the results. Considering this feedback, we recognize the importance of enhancing the control group in future research.
Originally, the control group had more cases, but after obtaining a detailed health interview and matching patients for age and body weight, only 29 cases remained.
In current paper, in the limitations of the study, we introduced the information, that due to a small number of control group our study should be treated as a preliminary.
- There is a problem with the IL-6 values between Tab. 1 and 2. In Tab. 1 IL-6 is in patients significantly higher, but the data in Tab. 2 do not show this accordingly.
According to point 2:
Yes, we apologize for our mistake and corrected it (Table 4). The range of IL-6 in the group of patients with RRMS was very wide (0.07-24.74 pg/mL).
The concentration of IL-6 was more similar in the case of division based on the applied therapy than based on gender. Furthermore, we checked other concentration/activity carefully.
Moreover, according to Reviewer’s#1 suggestion we also put the mean value and standard deviation in all tables.
- What about CSF data? Usually, CSF is available for MS patients. It would be more convincing for the reader if at least some of the results could be confirmed in CSF which is one of the diagnostic hallmarks of MS.
According to point 3:
We understand the significance of CSF data in diagnosing and understanding MS. CFS analysis is indeed one of the most diagnostic hallmarks of MS and can provide valuable insights into the disease process. While CSF data can offer valuable information, it requires an additional set of ethical considerations, specialized procedures, and a separate cohort of participants. Moreover, CSF analysis is an invasive investigation and obtain CSF from control group was impossible. Furthermore, we believe that the blood is easier and more possible biological fluid to observe any changes in pro/oxidant balance in the short period than CFS.
To include patients to Program of the Minister of Health the CSF was collected, but during regular follow-up appointments only blood was collected.
- Study limitations have not discussed in appropriate detail.
According to point 4:
Thank you very much for your valuable suggestion. At the end of the discussion section, we introduced new information (verse 453-474):
“There were several limitations to the current study. The investigation was performed in the blood not in CSF, which is one of the diagnostic hallmarks of MS. We acknowledge that including CSF data could potentially add further depth and confirmation to some of our findings. In future research, we would certainly consider expanding our investigation to incorporate CSF analysis, especially if the resources and the study design permit such an approach. Secondly, due to a small number of control group our study should be treated as a preliminary. Furthermore, it should be noted that there were differences in the number of patients, disease duration, and EDSS scores among the subgroups that received DMT. Therefore, the findings should be interpreted cautiously. Additionally, it is important to note that GPx is involved in various processes in the human body. For future studies, we plan to investigate the activity of glutathione reductase, glutathione S-transferase, as well as the concentrations of reduced and oxidized glutathione, selenium, and NADPH. Moreover, the investigation of genetic backgrounds is important in evaluating changes in the pro/antioxidant balance. This will allow us to further explore the potential causes of the decreased activity of GPx during RRMS.
In summary, the assessment of pro/antioxidant balance parameters indicates their significant alterations in RRMS patients. Specifically, there is a significant decrease GPx activity in patients with RRMS and increased CAT activity. We propose that the activity of GPx could be a promising marker and/or potential therapeutic target in RRMS. Further investigation is warranted to explore the potential factors contributing to drop in GPx activity.”

Round 2
Reviewer 1 Report
The authors have addressed my comments and suggestions and the revised manuscript sounds better.
Minor editing of English language required.
Reviewer 2 Report
My comments have been addressed by the authors accordingly.
None.